# Quantifying Functional Group Compositions of Household Fuel Burning Emissions

Emily Y. Li[1], Amir Yazdani[2], Ann M. Dillner[3], Guofeng Shen[1], Wyatt M. Champion[4], James J. Jetter[1], William T. Preston[5], Lynn M. Russell[6], Michael D. Hays[1], and Satoshi Takahama[2]

[1]Air Methods and Characterization Division, U.S. Environmental Protection Agency, Office of Research and Development, Research Triangle Park, North Carolina 27709, USA
[2]Laboratory for Atmospheric Processes and their Impacts, École Polytechnique Fédérale de Lausanne, CH-1015 Lausanne, Switzerland
[3]Air Quality Research Center, University of California, Davis, Davis, CA 95616, USA
[4]Air Methods and Characterization Division, Oak Ridge Institute for Science and Education, U.S. Environmental Protection Agency, Office of Research and Development, Research Triangle Park, North Carolina 27709, USA
[5]CSS Inc., Durham, North Carolina 27713, United States
[6]Scripps Institution of Oceanography, University of California San Diego, La Jolla, California 92093, USA

**Correspondence:** Emily Li (li.emily@epa.gov) and Satoshi Takahama (satoshi.takahama@epfl.ch)

**Abstract.** Globally, billions of people burn fuels indoors for cooking and heating, which contributes to millions of chronic illnesses and premature deaths annually. Additionally, residential burning contributes significantly to black carbon emissions, which have the highest global warming impacts after carbon dioxide and methane. In this study, we use Fourier transform infrared spectroscopy (FTIR) to analyze fine particulate emissions collected on Teflon membrane filters from fifteen cookstove types and five fuel types. Emissions from three fuel types (charcoal, kerosene, and red oak wood) were found to have enough FTIR spectral response for functional group (FG) analysis. We present distinct spectral profiles for particulate emissions of these three fuel types. We highlight the influential FGs constituting organic carbon (OC) using a multivariate statistical method and show that OC estimates by collocated FTIR and thermal optical transmittance (TOT) are highly correlated, with a coefficient of determination of $82.5\%$. As FTIR analysis is fast, non-destructive, and provides complementary FG information, the analysis method demonstrated herein can substantially reduce the need for thermal-optical measurements for source emissions.

## 1 Introduction

Residential burning is a major source of organic carbon (OC), and contributes approximately 30% of global emissions of black carbon, which is estimated to have the third highest global warming impact after carbon dioxide and methane (Ramanathan et al., 2008; Bond et al., 2013). The World Health Organization (WHO) estimates that close to 4 million premature deaths per year are associated with exposure to household air pollution, mainly from solid-fuel burning (WHO, 2014; Lee et al., 2020). Recent studies have typically categorized fine particulate matter ($PM_{2.5}$) emissions from household fuel burn emissions by mass, OC, and elemental carbon (EC) emission factors using gravimetric and thermal-optical methods (e.g., IMPROVE and NIOSH protocols) (Roden et al., 2006; Sharma and Jain, 2019; Jetter et al., 2012). However, thermal-optical OC-EC

measurements can sometimes take more than 45 minutes per sample and are destructive, while providing limited information, as OC and EC alone are only bulk properties of carbonaceous particles.

In contrast to thermal-optical methods, Fourier transform infrared spectroscopy (FTIR) measurements take a few minutes per sample, are non-destructive and highly reproducible, and provide more chemical information. FTIR measurements have been shown to be in good agreement with common methods for chemical characterization of organic matter (OM) such as aerosol mass spectrometry (AMS) and thermal-optical analysis (Takahama et al., 2013; Dillner and Takahama, 2015; Reggente et al., 2016; Takahama et al., 2016b; Yazdani et al., 2021a, b; Boris et al., 2019). However, FTIR has not been extensively used for chemical analysis of primary particulate emissions from residential burning.

In the current study, we use FTIR to quantify the organic functional group (FG) composition of $PM_{2.5}$ emissions from cook-stoves collected on polytetrafluoroethylene (PTFE) filters. We find that OC estimates from FTIR measurements and regression modeling are in good agreement with those from thermal optical transmittance (TOT). Our analysis highlights spectral differences among fuels (charcoal, kerosene, red oak wood, alcohol, and liquefied petroleum gas) and their particulate emissions during combustion. Estimates of aromatics and polycyclic aromatic hydrocarbons (PAHs) from FTIR measurements of particulate emissions from charcoal, kerosene, and red oak wood are compared with those using gas chromatography-mass spectrometry (GS-MS) measurements. Estimates of aromatics and PAHs from FTIR measurements have been previously unavailable for primary source emissions. Implications of this study include the widened and improved analytical-chemical portfolio for aerosol $PM_{2.5}$ emissions, leading to increased laboratory throughput and understanding of the compositional attributes of a known toxic $PM_{2.5}$ component.

## 2   Materials and Methods

An overview of cookstove testing, emissions sampling, OC-EC measurements, GC-MS PAH measurements, FTIR measurements, and post-processing are illustrated via flow chart in Fig. 1. Details of measurements and post-processing either described or cited from previous publications in the subsections below. In short, a range of stoves and fuels were tested for burn emissions, from which parallel PTFE membrane and quartz fiber (Qf) filter samples were collected. PTFE filters were used to measure gravimetric $PM_{2.5}$ mass, and then scanned using transmission-mode FTIR. The quartz filters were used to measure TOT OC-EC as well as GC-MS PAHs. From these sets of measurements, partial least squares regression (PLSR) models were built for the identification of important FTIR group frequencies for OC and PAH, and peak fitting and quantification of FGs, OC, and OM/OC were performed.

### 2.1   Cookstove emissions testing and sampling

Cookstove emissions tests were conducted at the U.S. Environmental Protection Agency cookstove test facility in Research Triangle Park, NC. Details of the facility and protocols are described in previous publications(Jetter and Kariher, 2009; Jetter et al., 2012; Shen et al., 2017a, b; Xie et al., 2018). We analyzed 152 samples from 15 different stoves and five fuel types,

pictured in Fig. 2. Fuels tested covered charcoal (56 samples), kerosene (21 samples), red oak wood (69 samples), alcohol (3 samples), and liquefied petroleum gas (LPG, 3 samples).

In this study, particulate emissions from stoves during the Water Boiling Test (WBT) were sampled. The WBT protocol (version 4, Global Alliance for Clean Cookstoves, 2014) includes three test phases: (1) a high-power cold-start (CS) phase in which the stove, pot, and water are at ambient temperature at the beginning of the test phase; (2) a high-power hot-start

(HS) in which the stove is hot at the beginning of the test phase; and (3) a low-power simmer phase (SIM) in which the water temperature is kept 3 °C below the boiling point (Sec. S1) (Jetter et al., 2012). A modified protocol described by Jetter et al. (2012) was used for the charcoal stove.

$PM_{2.5}$ was sampled isokinetically on Qf (47-mm diameter Tissuquartz™ pure quartz no binder, Pall Corporation) filters and PTFE membrane filters (47-mm diameter Teflo® membrane, Pall Corporation) positioned in parallel and downstream

of $PM_{2.5}$ cyclones (URG; Chapel Hill, NC) at a flow rate of 16.7 liters per minute. $PM_{2.5}$ was also sampled downstream of the PTFE filter using a Qf back filter for artifact correction. The ratio between OC on Qf back and front filters ranged between 0.06–1.79. The filter-based $PM_{2.5}$ sampling was conducted in a primary dilution tunnel (15.2 cm)for low-emission fuel-cookstove combinations (*e.g.*, forced-draft biomass stoves), and in a secondary dilution tunnel (25.4 cm) for those with high emissions (*e.g.*, natural-draft biomass stoves) to avoid overloading filters. Volumetric flow rates were $\sim$4.0 $\mathrm{m^3\,min^{-1}}$

and $\sim$26.8 $\mathrm{m^3\,min^{-1}}$ in the primary and secondary dilution tunnels, respectively. With the steady flow dilution tunnel system, dilution ratios vary as cookstove emissions fluctuate during testing. Temperatures at the filter sampling locations varied between (24–50°C) and (21–26°C) for the primary and secondary dilution tunnels, respectively. After sampling, filters were stored in filter petri dishes at -20°C to minimize volatilization and chemical reactions.

## 2.2 Measurements

From the PTFE and Qf filter samples collected, three quantitative measurements were obtained on all samples for use in FTIR post-processing: gravimetric $PM_{2.5}$ mass, TOT OC-EC measurements, and GC-MS PAH measurements. $PM_{2.5}$ mass was determined by weighing the PTFE filters (Pall Gelman) gravimetrically with a microbalance (MC5 Micro Balance, Sartorius) before and after sampling. FTIR spectra were gathered in transmission mode on the sampled PTFE filters. For FTIR analysis, the mid-infrared spectra in the 4000–400 $\mathrm{cm^{-1}}$ range were obtained using a Bruker-Vertex 80 FTIR instrument equipped with

an $\alpha$ deuterated lanthanum alanine doped triglycine sulfate (DLaTGS) detector, at a resolution of 4 $\mathrm{cm^{-1}}$, in transmission mode, averaged over 64 scans.

OC and EC on Qf samples were measured using a thermal-optical transmittance instrument (TOT; Sunset Laboratory) and a revised NIOSH method 5040 (NIOSH, 1999). Blank PTFE and Qf filters were also collected at the cookstove test facility and the associated $PM_{2.5}$ mass, OC, and EC backgrounds were measured and subtracted from test samples. In order

to make measurements on Qf (EC, OC, GC-MS) and PTFE (FTIR, gravimetric) filters more comparable (i.e., to minimize the differences due to volatilization and adsorption artifacts for semivolatile aerosols), artifact correction was performed. For this purpose, the values measured on the Qf back filters were subtracted from those on the Qf samples in parallel to the PTFE filter.

Thermal extraction GC-MS (TEx-GC-MS) as described by Shen et al. (2017b) was used to identify and quantify individual PAHs in sampled cookstove emissions for kerosene and red oak wood fuels. Briefly, a 0.4 cm$^2$ Qf punch (or three punches for artifact filters and low-emission samples from LPG) was placed inside a glass tube that was pre-baked to 550 °C and spiked with 1 μL of an internal standard mixture containing deuterated PAH compounds. Following automated tube insertion into the oven unit, the sample was heated to 325 °C at a rate of 50 °C/min and held for 11 min; 50 mL/min of helium (He) flowed over the sample. Desorbate was directed to a cryogenically cooled programmable temperature vaporization inlet at -85 °C operating in splitless mode. The GC inlet was rapidly heated at 720 °C/min to 325 °C. The sample was chromatographed on an ultralow-bleed capillary column (DB-5, 30 m × 0.25 mm I.D. × 0.25 μm film thickness) with He as the carrier gas (1 mL/min). The GC oven was temperature-programmed as follows: 65 °C for 10 min ramped to 325 °C at a rate of 10 °C/min and held for 15 min.

Chromatographed compounds (listed in Sec. S2) were detected using an Agilent 5973 MS detector operating in selected ion monitoring mode. The internal standard method and a multilevel, linear ($R^2 > 0.9$) calibration was used (0.1–1 ng) for PAH quantification. A mid-level calibration check was performed daily prior to sample analysis and was used as a continuing calibration in cases where measured and fixed concentrations were not within 20%. Daily checks were within 20% for most targets even when replicated over months. Mid-level check recoveries for PAHs were within $105\% \pm 33\%$ on average. PAHs below their detection limit were reported as "ND". Retention time shifts were negligible ($< 1\%$) throughout the analysis. Target analyte validation was achieved using the retention times and isotopic fragment ratios exceeding a 3:1 signal-noise ratio. Carryover tests were performed by reheating the TEx sample/apparatus immediately following the initial extraction and then examining the GC-MS chromatogram for the presence of target compounds, which indicated an extraction efficiency of $> 98\%$. Naphthalene was the only PAH detected ($0.16 \pm 0.11$ ng) above its method detection limit (0.02 ng) in the blank filters. Similar to EC and OC, artifact correction was performed for PAHs by subtracting values of the quartz back filters from those of the parallel filters.

## 2.3 Post-processing

The following subsections describe or cite from previous publications that describe the post-processing methods applied to FTIR spectra to identify and quantify FGs.

### 2.3.1 Baseline correction and blank subtraction

The FTIR spectra were baseline corrected using smoothing splines (Reinsch, 1967) to remove the contribution of light scattering by the PTFE filter membrane and particles, and absorption by carbonaceous material (Russo et al., 2014; Parks et al., 2019). For this work, the fitting procedure described by Kuzmiakova et al. (2016) was extended to wavenumbers below 1500 cm$^{-1}$ to access additional regions of the spectra. To reduce the influence of the PTFE absorption peak in the region 1300–1000 cm$^{-1}$, a scaled version of a baseline corrected blank filter spectrum was subtracted from each baseline corrected sample spectrum (Yazdani et al., 2021b). The blank spectrum used for subtraction was selected for each sample from a database of blank spectra (built from scans of similar batch of filters) using the closely matched intensity of the PTFE peak around 1150 cm$^{-1}$). The

final spectrum is the difference between the baseline corrected sample and the scaled baseline corrected blank spectra (Sec. S3). This procedure allowed us to identify and quantify some organic FG absorptions such as the out-of-plane (OOP) aromatic CH bands that are otherwise partially masked by the PTFE absorption bands.

Graphitic carbon contributes to the mid-infrared spectra through weak absorption bands of carbon bonds in aromatic rings near 1600–1500 $cm^{-1}$ (Tuinstra and Koenig, 1970; Friedel and Carlson, 1971; Țucureanu et al., 2016) and electronic transition absorptions (Russo et al., 2014; Parks et al., 2021). Both have been implicated in conferring capabilities for quantification of EC content by FTIR specrometry and statistical models in previous studies (Takahama et al., 2016b; Weakley et al., 2018; Parks et al., 2021), and our current analysis further supports the value of the electronic transition in quantification. The values of the smoothing spline baselines fitted to each sample spectra at 4000 $cm^{-1}$ was regressed against the magnitude of corresonding EC and OC loadings (major components of $PM_{2.5}$ in this study). A relatively large and statistically significant coefficient for EC (2.2 times the coefficient of OC, $R^2 = 0.72$; Sec. S4) suggests that EC contributes significantly to the FTIR baseline of combustion aerosols, and may be used for quantitative prediction of EC (Debus et al., 2021; Parks et al., 2021).

### 2.3.2   Quantification of functional groups

After determining the important FGs (by Variable Importance in Projection [VIP] scores method detailed in Secs. S5 and S6), their abundances were quantified using a multiple peak-fitting algorithm (Takahama et al., 2013; Reggente et al., 2019b). Aliphatic CH (aCH), aromatic CH (rCH), alcohol COH (aCOH), carboxylic acid (COOH), and non-acid carbonyl (naCO) groups were quantified in this work. OC concentrations and OM:OC ratios were then calculated using FG abundances with minimal assumptions about the number of carbon atoms attached to each FG (Takahama and Ruggeri, 2017).

For the calibration of aromatic CH groups using the OOP bending mode, we used the twelve samples of anthracene $C_{14}H_{10}$ analyzed by (Gilardoni et al., 2007) for the aromatic CH stretching mode. Anthracene powder (Sigma-Aldrich) was dissolved in high purity methanol (Fisher Scientific, Inc.), atomized with a constant output atomizer (TSI, Inc. model 3076) in nitrogen flow, and deposited on 37mm PTFE filters (Pall Gelman) restricted to a 1 $cm$ diameter collection spot with a custom-made PTFE mask (Gilardoni et al., 2007). Areal mass loadings for these samples spanned a range between 26.8 and 1040 $\mu g\,cm^{-2}$. The spectra aquired by FTIR were pretreated according Sec. 2.3.1 and the bending mode peak near 750 $cm^{-1}$ was used to build a calibration as described by Takahama et al. (2013) and Reggente et al. (2019a). In this case, the peak height is used rather than the peak area, and the calibration coefficient (interpreted as molar absorptivity) was estimated to be 0.017 $cm^2\,mol^{-1}$ of CH bond with a fit of $R^2 = 0.93$. The mass loading of PAHs was estimated from the quantified bonds assuming an effective C/H molar ratio of 1.4 (similar to anthracene) to account for additional carbon atoms of PAHs associated with the measured aromatic CH bonds.

### 3   Results and Discussion

In the following sub sections, we first present overall OC, EC, and $PM_{2.5}$ measurements among different fuel types, test phases, and stoves. Thereafter, we present FTIR spectra of particulate emissions qualitatively as well as those of unburned fuels. TOT

OC measurements are then combined statistically with the FTIR spectra to identify influential FGs for OC. Finally, influential FGs are quantified to estimate OC, OM, and OM/OC, as well as to compare with TOT OC and GC-MS PAH measurements.

## 3.1   OC, EC, and PM$_{2.5}$ emission factors

OC, EC, and PM$_{2.5}$ emission factors, expressed in grams of emission per MJ of energy delivered to the pot, are shown in Fig. 3. We observe a significant variation across red oak fueled stoves in terms of emission factors (Fig. 3). In particular, the 3-Stone Fire and the Envirofit G3300 stove (natural draft) have relatively high emission factors of OC and EC accompanied by lower modified combustion efficiencies (MCEs; Sec. S7). On the other hand, gasifier stoves such as EcoChula XXL have higher MCEs and around 10 times lower EC and OC emissions. Lower absorbances are also noticeable in the FTIR spectra of the EcoChula XXL stove especially in the CS and HS phases. OC and EC emission factors are slightly lower in the SIM phase compared to CS and HS phases. Nevertheless, the variations observed in the PM composition from red oak combustion across phases (characterized by elemental-carbon-to-total-carbon ratio, EC/TC, where TC = OC + EC) is not substantial when compared to the variability in each phase (Fig. 3 and Fig. S6). The average EC/TC for red oak combustion aerosols is $0.58 \pm 0.19$. This value is on the higher end of those reported in previous studies on the emission factors of particles emitted from wood-burning cookstoves (0.07–0.64) (Roden et al., 2006; Novakov et al., 2000).

Variation in emission factors across phases is more significant for charcoal combustion emissions compared to other fuels. The HS phase has the highest OC emission factor (Fig. 3). The EC emission factor and the EC/TC ratio increase significantly in the CS phase. The average EC/TC ratios of charcoal combustion aerosols for the CS phase and all phases combined are $0.78 \pm 0.23$ and $0.30 \pm 0.40$, respectively. Variability in results was observed in charcoal combustion tests due to non-uniform fuel and differences in combustion conditions. Lump charcoal (not briquettes) was used for testing because it was representative of fuel usually used in low- to middle-income countries. Charcoal had non-uniform size and shape, and it was screened between 2.5 and 5.0 cm. Variation in the fuel bed packing due to the non-uniform pieces and variation in the carbonization of the charcoal can affect combustion and emissions. Kerosene was used as an accelerant to ignite the charcoal for all cold-start test phases, with a small amount of wood kindling used for the EcoZoom stove only, where red oak was ~9% of total fuel mass consumed. No accelerant was used during hot-start or simmer test phases. Different combustion characteristics between stoves strongly influences emissions. For example, turbulence/mixing within the combustion chamber of a traditional charcoal stove is a key factor in devolatilization of fuel and formation of organic PM (Lea-Langton et al.). Field testing of traditional charcoal stoves finds relatively steady emissions of CO (and hence modified combustion efficiency) during the entire test period, while contributions of PM$_{2.5}$, black carbon (BC), and light scattering (B$_{sp}$, $\lambda = 635$ nm, an optical-based proxy for EC) largely occur during the ignition period (e.g., 62% and 67% of BC and B$_{sp}$ respectively occurred during the first 20% of testing) (Champion and Grieshop). Therefore, both particle composition and emission rate change drastically during charcoal ignition, indicative of the highly variable process.

## 3.2 FTIR spectra and interpretation

Figure 4 exhibits distinct spectral profiles for PM$_{2.5}$ emitted from combustion of different fuel types. The FTIR spectra of alcohol and LPG particulate emissions were omitted as they were noisy and had organic absorptions comparable to those of blank filters. This observation is corroborated by the negative or close-to-zero TOT OC measurements for alcohol and LPG emissions.

  The spectra of red oak combustion aerosols display a strong and broad band at 3500 cm$^{-1}$ related to aCOH across all phases.
This group is abundant in wood constituents (lignin, cellulose, and hemicellulose). In addition, medium aCH absorbances at 3000–2800 cm$^{-1}$, medium carbonyl CO absorbances at around 1700 cm$^{-1}$, and strong aromatic C=C absorbances at 1600 cm$^{-1}$ are observed in the spectra of red oak combustion aerosols. The absorption band attributed to the lignin aromatic ring stretching vibration at 1515 cm$^{-1}$ is also prominent in several spectra (Boeriu et al., 2004), consistent with the existence of aromatic compounds with similar structure to monolignols, which are found in plants. Aromatic CH OOP bands at around 750
190 cm$^{-1}$ (Centrone et al., 2005) and occasionally aromatic stretching aCH bands at around 3050 cm$^{-1}$ are also visible in the FTIR spectra. The mean spectrum of the SIM phase has the weakest aromatic OOP band among all phases. Levoglucosan signatures described by Yazdani et al. (2021b) are observed in several spectra for all three phases. Inorganic nitrate absorbances at 1400 and 830 cm$^{-1}$ are visible in some red oak samples especially in CS and HS phases.

  The mid-infrared spectrum of unburned kerosene fuel (from Xia et al., 2017) displays strong aCH peaks and weak rCH
stretching (3050 cm$^{-1}$) absorbances. This is consistent with kerosene fuel being predominantly composed of hydrocarbons, the majority of which are straight-chain and cyclic with the rest being aromatic (Collins, 2007). An elemental analysis of unburned kerosene fuel (Sec. S8) also shows the insignificant contribution of elements other than H and C and a molar H/C ratio of about 2:1 suggesting long-chain or cyclic alkanes.

  In contrast, aCH absorbances are much weaker in the particulate emissions of kerosene combustion in all three phases, with
200 high CH$_3$/CH$_2$ peak ratios that indicate branched or small hydrocarbons. Furthermore, rCH absorptions are more prominent in the particulate emissions, suggesting a significant formation of aromatic hydrocarbons and PAHs during the combustion. In the majority of spectra, moderate aCOH absorbances are observed. In several samples taken from the emissions of the kerosene wick-type cookstove, a sharp CO carbonyl peak at 1695 cm$^{-1}$ is observed (Sec. S9).This peak is accompanied by a broad peak at 3400–2400 cm$^{-1}$ (with doublets at 2600 and 2400 cm$^{-1}$ attributed to the dimerized carboxylic acid OH), suggesting high
abundances of carboxylic acids (Pavia et al., 2008). The carbonyl peak with a relatively low frequency and the high abundance of rCH indicate the existence of aromatic acids. Aromatic signatures appear to be more prominent in the CS and HS phases.

  The charcoal spectrum (from Guo and Bustin, 1998) has weak aCH and carbonyl CO absorbances, and strong aromatic C=C and aCOH absorbances. The weak aCH peaks are consistent with the low hydrogen content of the fuel (Sec. S8).However, the mid-infrared spectra of particulate emissions from charcoal combustion show strong aCH peaks (Fig. 4). The aCH peaks are
210 especially prominent in the HS phase in which charcoal is added to the hot fuel bed (Fig. S1). In addition, the carbonyl CO peak at 1700 cm$^{-1}$ emerges in the charcoal combustion spectra in the HS phase, while being absent in the charcoal spectrum itself. This observation suggests the formation of carbonyls during charcoal pyrolysis and burning. Charcoal combustion in the

cold start (CS) phase has the highest EC emission factor (Fig. 3), concurrent with prominent C=C absorbances at around 1600 $cm^{-1}$ in the FTIR spectra. The ratio of the aromatic C=C peak (1600 $cm^{-1}$) to aromatic CH (750 $cm^{-1}$) is higher for charcoal burning aerosols compared to other fuels, suggesting that aromatic compounds are poor in hydrogen (=C–H groups).

### 3.3 Quantified functional groups from FTIR spectra

The organic functional groups aliphatic CH (aCH), aromatic CH (rCH), alcohol COH (aCOH), carboxylic acid (COOH), and non-acid carbonyl (naCO) groups were quantified by peak fitting to mid-infrared. The results of peak fitting are shown in Fig. 5 in terms of OM emission factors and OM/OC ratios (FG emission factor values and corresponding absorption bands are tabularized in Sec. S10). OC estimated by FTIR from the abundances of FGs ("FG OC") is highly correlated with TOT OC measurements ($R^2 = 0.83$; Fig. S14). However, FG OC underpredicts TOT OC by around 40%, although artifact correction has been performed for TOT OC using Qf back filters. This underprediction can be attributed to imperfect correction of quartz adsorption artifacts, volatilization of compounds from particles on PTFE filters, uncharacterized FGs, uncertainty in FG absorption coefficients, underprediction of the fractional carbon associated with each FG, and the operationally-defined EC and OC separation threshold for TOT. Volatilization losses off filters (following a denuder) for urban ambient samples have been reported to be on the order of 10% (Subramanian et al., 2004), though the amount may be different for fresh emissions that can contain substances that can revolatilize (Robinson et al., 2007). Underprediction in OC due to uncharacterized FGs is reported to be on the order of 8–20% considering the range of structures in compounds expected in OM (Takahama and Ruggeri, 2017). OC estimates by TOT have been reported to be higher than that by reflectance correction by 0–35% due to charring profile through the filter (Chow et al., 2004; Chiappini et al., 2014). The magnitude of each contribution across samples can vary substantially. A combination of these factors can lead to the overall discrepancy observed in this study, which is on the higher end relative to comparisons of FTIR OC or OM to different techniques in previous studies (e.g., Liu et al., 2009; Russell et al., 2009; Hawkins and Russell, 2010; Takahama et al., 2011; Corrigan et al., 2013; Liu et al., 2018; Reggente et al., 2019a). Nonetheless, the consistent mass recovery permits us to make systematic comparisons regarding the OM composition using FG analysis.

The concentration of rCH was estimated in this work using the OOP band (750 $cm^{-1}$), which is more distinguishable than the stretching mode vibration (3050 $cm^{-1}$) previously used. The OOP peak has a comparatively higher absorption coefficient and does not overlap significantly with other group frequencies as does the stretching mode peak (e.g., with the ammonium NH stretch). Peaks of the two vibrational modes exhibits a strong relationship (Fig. S11), but the OOP peak is expected to have higher sensitivity due to the aforementioned reasons. Contrary to the peak at 3050 $cm^{-1}$, the peak at 1600 $cm^{-1}$ attributed to aromatic C=C is not highly correlated with the OOP peak (Fig. S12), especially for red oak burning aerosols — likely due to high abundance of lignin pyrolysis products substituted rings. However, a qualitative relationship between rCH and TOT EC concentrations, dependent on fuel type, can be observed (Fig. S13).

The estimates of rCH using this OOP bend are highly correlated ($R^2 = 0.84$; slope = 19.6) with the total PAH concentration by GC-MS measured for kerosene and red oak wood fuels (Fig. 6). The VIP scores of GC-MS total PAH concentration regressed against mid-infrared absorbances also indicate the importance of the 750 $cm^{-1}$ band for PAHs (Fig. 7). The C=C

band near 1500-1600 $cm^{-1}$ is also indicated as a predictive feature, but exhibits less selectivity toward PAHs since contributions from other aromatic structures present such as graphitic carbon and lignin are also present. The magnitude of PAHs and aromatic estimated using the OOP band at 750 $cm^{-1}$ is almost an order of magnitude higher than that measured by GC-MS probably due to this band accounting for other unmeasured aromatic and PAH compounds. While our calibration for the total PAH is based on a single compound, variations in absorption coefficients on the order of 30% for the same FGs across different compounds have been reported (Takahama et al., 2013; Reggente et al., 2019a). Furthermore, the C/H ratio assumed (1.4) for calculation of PAH concentrations by FTIR (Sec. 2.3.2) is a conservative one considering that the range of C/H values for compounds measured by GC-MS in this study was 1.0–2.0 (Sec. S2), and the highest abundance found for compounds with C/H greater than 1.44 (e.g., pyrene and fluoroanthene, benzo-a-pyrene). Adoption of higher values for this ratio leads to higher estimates of PAH, OM mass, and proportion of PAH in estimated OM; the value of this and the absorption coefficient can be further constrained in future studies with additional experiments and inverse modeling approaches (e.g., Bürki et al., 2020). In summary, the apparent difference in magnitude of FTIR and GC-MS estimates of PAH abundance are beyond the levels of our anticipated uncertainty, and likely stems from the difference in approach between FG and molecular level analysis toward quantification of mass concentrations (Russell, 2003).

Summarizing the FTIR composition analysis, red oak combustion aerosols have the highest mean OM emissions factors for all phases (Fig. 5), consistent with TOT OC measurements (Fig. 3). For all three phases, aCOH, aCH, and rCH groups constitute the majority of OM. Among different phases, the simmering (SIM) phase has a slightly lower OM emission. The rCH group abundance is significantly lower in the SIM phase. The lower concentration of this group in the SIM phase is also observed for other fuel types. This is also reflected in the higher OM/OC ratio of this phase compared to other phases. The average OM/OC ratio of red oak combustion aerosols is estimated to be around 1.7.

Kerosene combustion has the lowest average OM emissions factors compared to other fuels in all phases (Fig. 5; around 4 times lower than red oak). Interestingly, while kerosene OC and EC is not necessarily higher than others (Fig. 3), its proportion of aromatic CH (rCH) is higher (Fig. 5). No clear difference is observed between the mean OM emission factor among different phases. The rCH is estimated to be the most prominent FG for the kerosene burning aerosols followed in order by aCOH, aCH, and COOH. The average OM/OC ratio is 1.8 for the kerosene aerosols. Like red oak, the SIM phase has the highest OM/OC ratio due to a higher abundance of COOH (more oxygenated) and lower abundance of rCH (likely due to more complete combustion).

Charcoal burning aerosols have average OM emission factors slightly lower than those of red oak combustion (Fig. 5). The aCH group is the most important FG for charcoal burning aerosols constituting up to 50% of OM. The aCOH and rCH are the most abundant groups after aCH. Other FGs exist in negligible amounts in charcoal burning aerosols. The average OM/OC ratio for charcoal burning aerosols is 1.6, slightly lower than that of other fuels. Like other fuel types, the SIM phase has the highest OM/OC ratio and the lowest abundance of aromatics.

There is a dearth of estimates of OM/OC ratios from cookstove emissions for direct comparison. The similarity in the range of OM/OC ratios (1.6–1.8) between fossil fuel and red oak combustion in this work stand in slight contrast to differences observed in a previous study comparing primary OM from coal combustion and wood burning in furnaces where average

OM/OC ratios were 1.4 and 1.6, respectively, which were similar to AMS (Yazdani et al., 2021b). The current values are higher than reported by AMS for submicron HOA (1.3–1.5), though closer to the range of laboratory-generated and inverse-modeled BBOA (>1.5) Canagaratna et al. (2015). OM/OC estimates from FTIR can be biased high if some unfunctionalized carbon is not considered in the calculation (Takahama and Ruggeri, 2017; Reggente et al., 2019a); in that case the OM/OC would reflect the ratio for polyfunctional carbon atoms that are extracted by our current FTIR calibrations. In the study by Yazdani et al. (2022), the FTIR OM was estimated to be 30% higher than AMS measurements without collection efficiency correction, while FTIR OC is underestimated by about 40% compared to TOT OC in this work. Bürki et al. (2020) proposed an approach to find parameters for estimation of FTIR OC (including calibration coefficients and undetected carbon fraction) that were most consistent with the observed TOR OC concentrations in collocated ambient measurements. These model parameters were then used to obtain revised estimates of OM/OC, which resulted in lower OM/OC estimates as the original FTIR OC estimates underpredicted the TOR OC concentrations by approximately 40%. Such an approach can also be considered for further investigation of cookstove primary emissions. Nonetheless, the bias of TOT OC is systematically similar across fuel types in this work, so the relative differences among them reported here are likely to be consistent with respect to this source of bias. Differences in absorptivities for the same functional group found in different compounds across sources may exist, but are not accounted for in this study. More studies in this area are warranted given the difficulty in quantifying OM/OC using various techniques, and lack of definitive reference methods. Molecular methods sample a small subset of molecules present (Rogge et al., 1993), AMS relies on calibrations to a set of selected representative compounds to adjust for ionization losses (Aiken et al., 2007; Canagaratna et al., 2015), reconstructed fine mass regression are subject to compounding analytical errors (Hand et al., 2019), and estimation of OM through thermo-gravimetric analysis (Polidori et al., 2008) is prohibitively labor-intensive (and is also subject to compounding analytical errors).

Previous studies correlate aromatics and PAHs with mutagenicity and carcinogenicity (Riedel et al., 2018; Shiraiwa et al., 2017; Gibbs-Flournoy et al., 2018; Mutlu Esra et al., 2016). Our analysis shows that these compounds are abundant even in relatively clean fuels (*i.e.*, with low emission factors) like kerosene, which has a magnitude of emitted aerosol PAH as high as charcoal, suggesting its harmfulness. This observation along with the highly oxygenated FGs present in kerosene OM is consistent with previous studies providing some evidence that kerosene emissions may impair lung function and increase infectious illness, asthma, and cancer risks (Lam et al., 2012).

## 4 Conclusions

This manuscript introduces FTIR as a measurement technique that can provide new insights into the organic composition of aerosol emissions from cookstoves, in which variations in composition and overall emissions due to fuel type and operational phase of the cookstove are evaluated. The FTIR analysis is applied directly to filters collected for gravimetric mass analysis and does not require an additional sampling line. The OM composition in terms of oxygenated functional groups - carboxylic acids, carbonyls, and alcohols - and PAHs vary substantially across fuel types, and in some cases, the operational phase of the stove. Additionally, a qualitative comparison of emitted OM to the composition of the fuel is demonstrated through comparison of the

aerosol spectra to the spectra of the unburnt fuel. The overall organic carbon estimated by FTIR is approximately 60% of that estimated by thermal optical transmittance, with an $R^2$ of 0.82. This mass recovery is on the low end of previously reported OC and OM comparisons against thermal optical methods or aerosol mass spectrometry, and may indicate the additional sampling artifacts in emissions studies, or the greater presence of carbon associated with uncharacterized functional groups or skeletal carbon atoms that are missed by FTIR. A new development introduced in this manuscript is the quantification of

PAH concentrations — particularly important in fresh emissions — based on the OOP bending absorption peak of aromatics by FTIR, which is more prominent than the stretching vibrations observed in the functional group region of the spectrum. The estimate of PAH hydrocarbon mass loadings by FTIR is found to be five-fold greater than that the summed from 16 PAHs analyzed by GC-MS, likely because FTIR estimates account for unmeasured PAH compounds in the samples. Thus, FTIR can potentially provide an additional dimension of information for toxicity studies of emitted aerosols.

*Author contributions.* EYL, AMD, MDH, and ST conceived of the project and manuscript. GS performed the filter sample collection and gravimetric analysis, WTP directed the OC-EC and PAH measurements and analysis, and EYL performed the FTIR measurements. EYL, GS, WMC, JJJ, WTP, and ST contributed further data curation. AMD, JJJ, MDH, LMR, and ST provided resources and/or acquiring financial support for the project. EYL, AY, and ST provided and implemented code and supporting algorithms and code components, and prepared data visualization. EYL, AMD, MDH, and ST provided oversight and leadership responsibility for research planning and execution, including

mentorship. EYL, AY, WMC, MDH, and ST contributed to the initial draft, and all authors contributed to the review and editing of the written manuscript.

*Competing interests.* The authors state no competing interests.

*Disclaimer.* The views expressed in this article/chapter are those of the authors and do not necessarily represent the views or policies of the U.S. Environmental Protection Agency. Mention of trade names or commercial products does not constitute endorsement or recommendation

for use.

*Acknowledgements.* The authors thank the U.S. Environmental Protection Agency Office of Air Quality Planning and Standards for their administrative support, the Office of Research and Development for Quality Assurance support, and acknowledge funding from the Swiss National Science Foundation (200021_172923).

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

# Figures

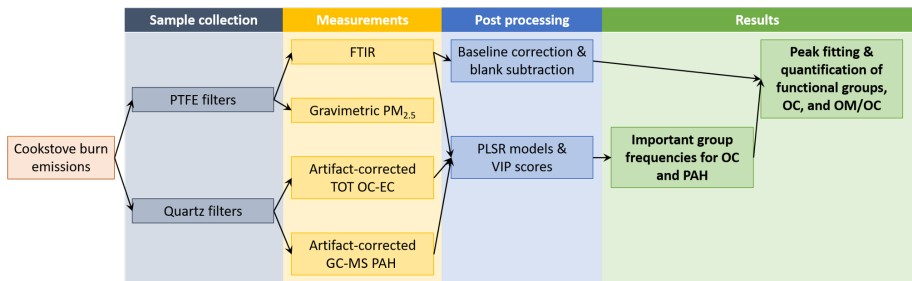

**Figure 1.** Flow chart from sample collection to measurements, post-processing, and results.

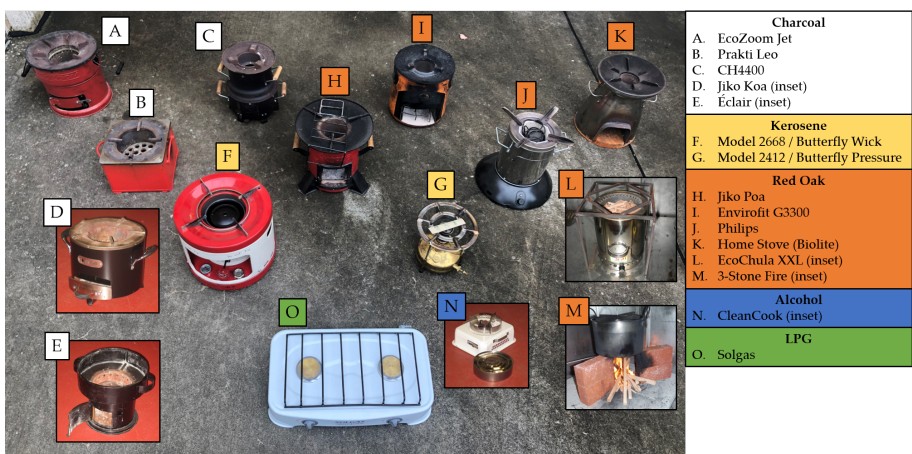

**Figure 2.** Photo showing 15 stoves used in this work. Charcoal stoves: (A) EcoZoom Jet, (B) Prakti Leo, (C) Envirofit CH4400, (D) Jiko Koa (inset), and (E) Éclair (inset). Kerosene stoves: (F) Model 2668 / Butterfly Wick and (G) Model 2412 / Butterfly Pressure. Red oak wood stoves: (H) Jiko Poa, (I) Envirofit G3300, (J) Philips HD4012, (K) Home Stove (Biolite), (L) EcoChula XXL (inset), and (M) 3-Stone Fire (inset). Alcohol stove: (N) CleanCook (inset). LPG stove: (O) Solgas.

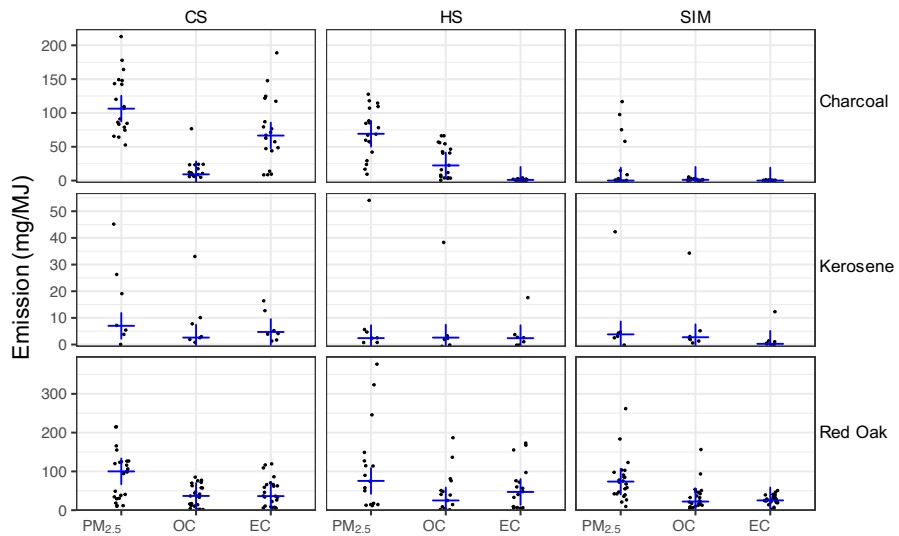

**Figure 3.** Emission factors $(\mathrm{mg/MJ})$ of gravimetric $PM_{2.5}$ and artifact-corrected TOT OC and EC, separated by fuel type and test phase (CS: cold start, HS: hot start, SIM: simmering). Blue crosses show the median for each category.

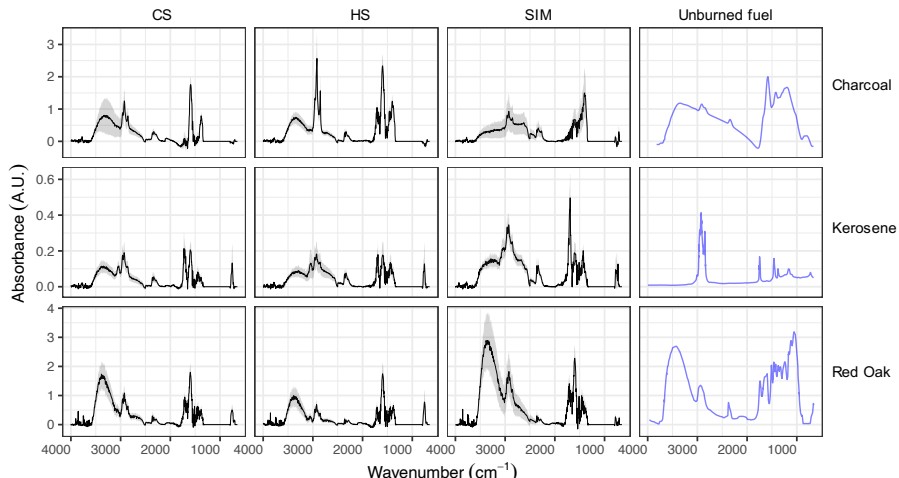

**Figure 4.** Average mid-infrared spectra of unburned fuels and their particulate emissions separated by source and test phase. Emission spectra are shown in terms of emission factors (absorbance divided by MJ energy delivered to the pot). Shaded bands show the mean spectrum $\pm$ 0.5 standard deviation. CS = cold start, HS = hot start, SIM = simmering. Features in the region betwen 1000–1300 $\mathrm{cm}^{-1}$, which has interferences from PTFE filters, has been omitted. Spectra from unburned fuels (blue lines), presented with arbitrary scaling, are taken from the literature: Guo and Bustin (1998) for charcoal, Xia et al. (2017) for kerosene, and Pandey (1999) for red oak.

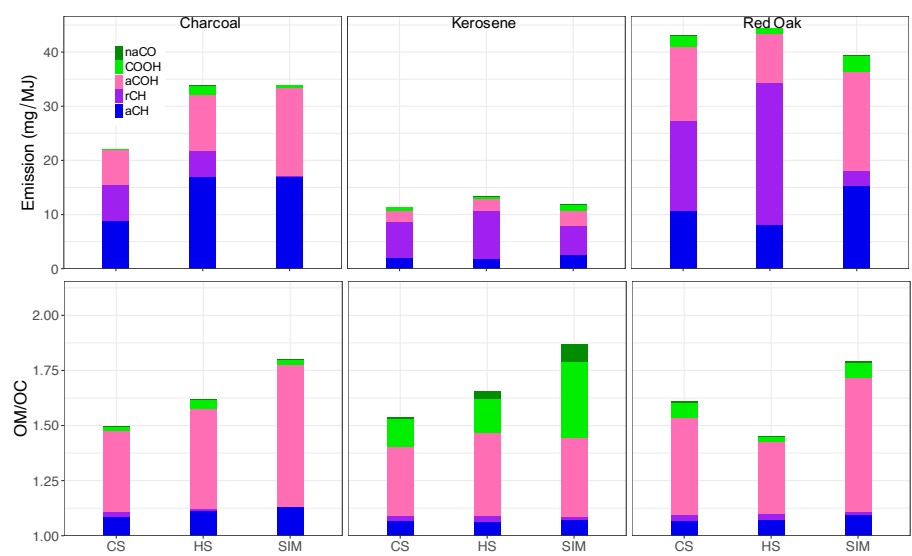

**Figure 5.** (Top row) OM emission factors calculated from mid-infrared spectroscopy, separated by functional group contribution, and averaged over each phase. (Bottom row) OM/OC ratios calculated from mid-infrared spectroscopy, separated by functional group contribution. CS = cold start, HS = hot start, SIM = simmering.

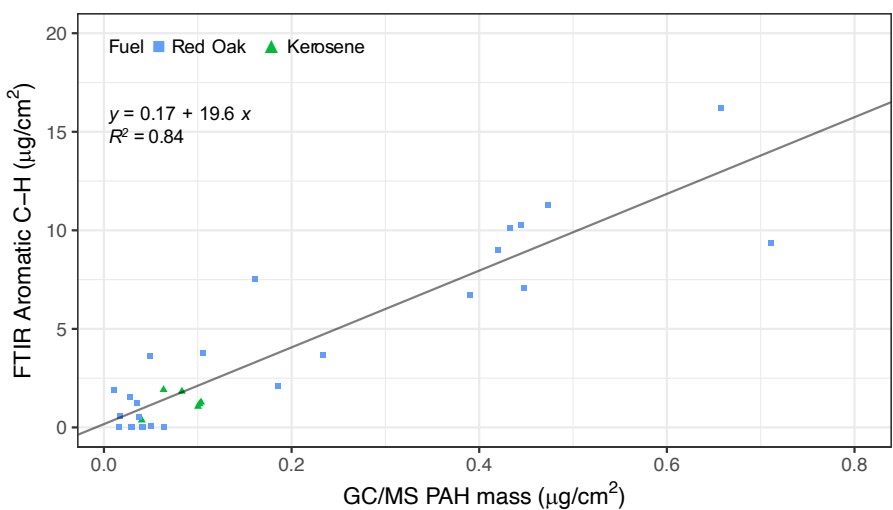

**Figure 6.** Scatter plot of aromatic CH concentration estimated using the peak at $750 \text{ cm}^{-1}$ and GC-MS total PAH concentration.

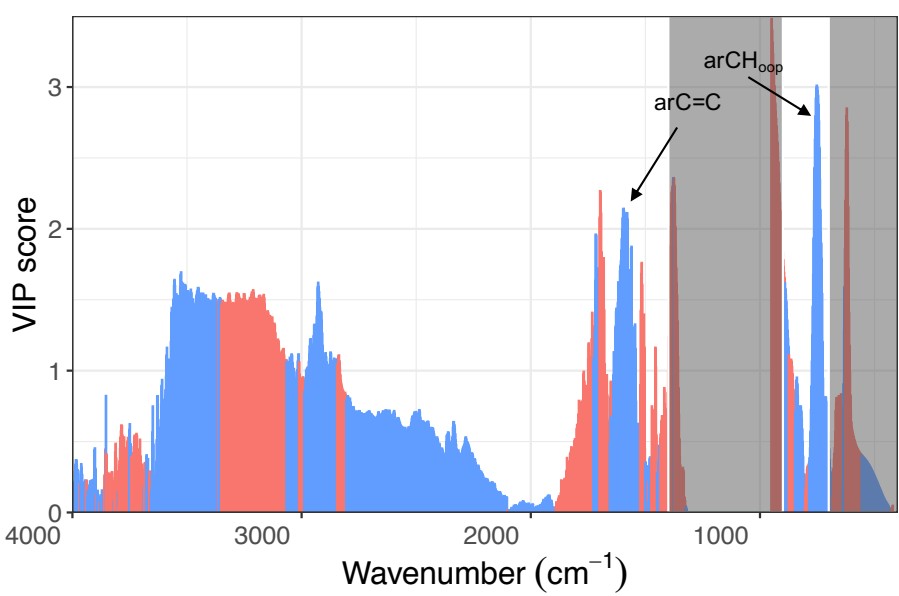

**Figure 7.** VIP scores of GC-MS sum of PAHs regressed against FTIR spectra.