# Peer review of "Quantifying Functional Group Compositions of Household Fuel Burning Emissions"

_Atmospheric Measurement Techniques, 2023_

## Author Comment (AC2)

**Response to reviewers for**
**"Quantifying Functional Group Compositions of Household Fuel Burning Emissions" by Li et al. (amt-2023-90)**

**Reviewer 1**

The manuscript entitle "Quantifying Functional Group Compositions of Household Fuel Burning Emissions" describes the application of FTIR analysis for the identification of organic functional group (OFG) in particulate matter emitted during combustion of different fuels by household appliances. In addition, different combustion phases are investigated. The results obtained by FTIR analysis are compared to those obtained from traditional analytical techniques, including thermal-optical analysis (for total OC determination) and GC-MS (for the quantification of PAHs).

The manuscript presents new data on OFG composition of primary emissions and describes the value of FTIR measurements to investigate the aromatic aerosol component, which is extremely relevant for defining the impact of combustion aerosols on human health. Some more discussion would be useful to understand the discrepancies between thermal optical OC and FTIR OC.

The manuscript is clear and well written. I recommend its publication after minor revision.

We thank the reviewer for the encouraging evaluation.

Specific comments.

1. Section 2.1 Can the authors add any details about the dilution ratio of the sampled emissions and the temperature at sampling point?

   The following text has been included:

   "Volumetric flow rates were $\sim$4.0 m$^3$ min$^{-1}$ and $\sim$26.8 m$^3$ min$^{-1}$ in the primary (6") and secondary (10") dilution tunnels, respectively. With the steady flow dilution tunnel system, dilution ratios vary as cookstove emissions fluctuate during testing. Temperatures at the filter sampling locations varied between (24–50°C) and (21–26°C) for the primary and secondary dilution tunnels, respectively."

2. Line 78 What is the ratio between the OC quartz back filter and the OC measured on the Qf collected in parallel?

   The following text has been added:

   "The ratio between OC on Qf back and front filters ranged between 0.06–1.79."

   This ratio is known to range widely, with lower ratios for increased loading as the filter saturates.

3. Line 198: As suggested by the authors, one of the reason for the underestimation of FTIR-OC compared to TOT-OC is the "operationally defined EC-OC separation". Do the authors observed a link between the OC underestimation and the pyrolitic carbon quantified by thermal-optical analysis?

The bias between FTIR OC and TOT OC over remains indiscernable (within a wide range of variation) above pyrolitic carbon ("PyC") loadings of 5 $\mu g/cm^2$, while PyC to TOT OC ratio by 10% on average over this same range. At low PyC concentrations, PM loadings are low overall and FTIR OC likely exhibits a high relative bias due to PTFE interferences, though this variation does not preclude effects of pyrolysis.

[Figure]

It may be that the optical transmittance correction captures and corrects for the relative trend in pyrolized organic fraction, leading to a consistent relationship between FTIR OC and TOT OC. Another perspective on the bias of TOT OC measurements due to the operational definition in EC-OC separation is provided through comparison with thermal optical reflectance (TOR) in other studies. For instance, Chiappini et al. (2014) report that OC estimates can diverge between these two methods, with TOT OC systematically higher than TOR OC. A more comprehensive discussion in the text is included in response to Reviewer 2, Comment #3.

Chiappini, L., Verlhac, S., Aujay, R., Maenhaut, W., Putaud, J. P., Sciare, J., Jaffrezo, J. L., Liousse, C., Galy-Lacaux, C., Alleman, L. Y., Panteliadis, P., Leoz, E., and Favez, O., "Clues for a standardised thermal-optical protocol for the assessment of organic and elemental carbon within ambient air particulate matter," English, *Atmospheric Measurement Techniques*, vol. 7, no. 6, pp. 1649–1661, 2014, Publisher: Copernicus GmbH. doi:10.5194/amt-7-1649-2014.

4. In addition, is it possible that the sensitivity of FTIR is reduced by the signal attenuation due to high EC loading? The agreement between TOC-OC and FTIR-OC is generally higher in ambient samples, where I assume the OC to EC ratio is higher. Do you see a link between the underestimation of FTIR OC and the OC to EC ratio?

High EC loading does not lead to substantial attenuation of the signal since the electronic transition is small compared to the scattering contribution from PTFE (Mcclenny et al., 1985; Parks et al., 2021) and absorption bands of graphitic carbon defects (below 1600 $cm^{-1}$) apparent in the infrared spectrum are relatively weak (discussed by Friedel and Carlson, 1971; Takahama et al., 2019).

Friedel, R. A. and Carlson, G. L., "Infrared spectra of ground graphite," *The Journal of Physical Chemistry*, vol. 75, no. 8, pp. 1149–1151, 1971. doi:10.1021/j100678a021.

Mcclenny, W. A., Childers, J. W., Rōhl, R., and Palmer, R. A., "FTIR transmission spectrometry for the nondestructive determination of ammonium and sulfate in ambient aerosols collected on teflon filters," *Atmospheric Environment (1967)*, vol. 19, no. 11, pp. 1891–1898, 1985. doi:10.1016/0004-6981(85)90014-9.

Parks, D. A., Griffiths, P. R., Weakley, A. T., and Miller, A. L., "Quantifying elemental and organic carbon in diesel particulate matter by mid-infrared spectrometry," *Aerosol Science and Technology*, vol. 0, no. 0, pp. 1–14, 2021. doi:10.1080/02786826.2021.1917764.

Takahama, S., Dillner, A. M., Weakley, A. T., Reggente, M., Bürki, C., Lbadaoui-Darvas, M., Debus, B., Kuzmiakova, A., and Wexler, A. S., "Atmospheric particulate matter characterization by Fourier transform infrared spectroscopy: A review of statistical calibration strategies for carbonaceous aerosol quantification in US measurement networks," *Atmospheric Measurement Techniques*, vol. 12, no. 1, pp. 525–567, 2019. doi:10.5194/amt-12-525-2019.

5. Line 209-202: Can the authors comment on the potential artefact of soot/graphitic carbon/EC on the aromatic CH signal? (Fig. S12) The conclusion about the contribution of multiple PAH, in addition to those quantified by GC-MS is convincing. Nevertheless, one of the strongest points of the manuscript is the ability of FTIR measurements to describe the totality of the aromatic component. So it would be good if the authors could say something about the potential artefacts on the PAH quantification due to EC.

The out-of-plane (OOP) aromatic CH is specific to the C=C-H bonds in PAHs, which are absent in graphitic carbon. Graphitic carbon and PAHs exhibit weak intensity bands near 1600–1500 cm$^{-1}$ for C=C bond stretching which have been used previously for EC quantification (also now discussed in the main text in response to Reviewer 2, Comment #1), but we do not use this band in this work. Soot and EC are thought to contain low-volatility organic compounds (e.g., Chow et al., 2004; Lack et al., 2014), but this OOP aromatic CH band is distinct from other organic bands such as aliphatic or alkene CH and oxygenated groups.

Chow, J. C., Watson, J. G., Chen, L.-W. A., Arnott, W. P., Moosmüller, H., and Fung, K., "Equivalence of Elemental Carbon by Thermal/Optical Reflectance and Transmittance with Different Temperature Protocols," *Environmental Science & Technology*, vol. 38, no. 16, pp. 4414–4422, 2004. doi:10.1021/es034936u.

Lack, D. A., Moosmueller, H., McMeeking, G. R., Chakrabarty, R. K., and Baumgardner, D., "Characterizing elemental, equivalent black, and refractory black carbon aerosol particles: A review of techniques, their limitations and uncertainties," *Analytical and Bioanalytical Chemistry*, vol. 406, no. 1, pp. 99–122, 2014. doi:10.1007/s00216-013-7402-3.

Technical corrections

1. Several citations are reported without leaving a space before the brackets. For ex. lines 13 and 15.

2. Line 194: S12 should be S13

We have corrected the Supplement so that Figure S11 has been removed and this error is corrected.

3. Line 197: I guess the term "variability" would be more accurate than "uncertainty"

The reviewer is correct in that there is variability in absorption coefficients for the same functional group in different molecules that can lead to biases in our quantification. However, we do mean "uncertainty" in the sense that we do not know what the value for the (effective) absorption coefficient that is most appropriate for these particular samples.

4. Line 208 : S11 should be S12

   We have corrected the Supplement so that Figure S11 has been removed and this error is corrected.

5. Fig S11 and fig 7 looks the same. Please remove one of them

   We thank the error for catching this error. We have corrected the Supplement so that Figure S11 has been removed.

**Reviewer 2**

The manuscript titled "Quantifying Functional Group Compositions of Household Fuel Burning Emissions" by Li et al. discusses the utilization of Fourier transform infrared spectroscopy (FTIR) to analyze fine particulate emissions originating from various cookstoves. The study presents source profiles of functional groups derived from different fuel types and cookstoves. Quantitative outcomes were achieved by comparing the results with OC/EC and GC-MS measurements. Overall, the manuscript is well-written and aligns with the scope of AMT. I recommend its publication after some minor revisions.

We thank the reviewer for the encouraging evaluation.

Minor aspects

1. The authors need to clearly state how the baseline is determined. Which is to say whether the baseline is defined by individual Teflon filters before sampling, or a unified baseline is used. It would be important for the researchers who would like to follow this method.

   We have rewritten the section on baseline correction (2.3.1) and added a section to the supplement (S3) with illustrations.

2. It would greatly benefit readers to include an example plot in the supplementary information illustrating the subtraction process of both blank baseline and EC-influenced baseline.

   We have rewritten the section on baseline correction (2.3.1) and added a section to the supplement (S3) with illustrations.

3. Line 149, it would be valuable to provide insights into the factors contributing to the substantial variability observed in charcoal combustion tests. Possible connections with temperature or other combustion conditions could be explored and explained.

   We have added this discussion in the main text:

[revised manuscript text omitted]

6. I recommend including a table summarizing the functional group abundances alongside their corresponding typical wavenumbers from source profiles of different cookstoves. This addition will be immensely helpful for future researchers interested in employing the same method.

We have included such a table in the new Section S10.